# Investigation of Impact of Walking Speed on Forces Acting on a Foot–Ground Unit

**DOI:** 10.3390/s22083098

**Published:** 2022-04-18

**Authors:** Barbara Jasiewicz, Ewa Klimiec, Piotr Guzdek, Grzegorz Kołaszczyński, Jacek Piekarski, Krzysztof Zaraska, Tomasz Potaczek

**Affiliations:** 1Department of Orthopedics and Rehabilitation, Medical College, Jagiellonian University, Balzera 15, 34-500 Zakopane, Poland; tomaszpotaczek@gmail.com; 2Łukasiewicz Research Network—Institute of Microelectronics and Photonics, Kraków Division, Zablocie 39, 30-701 Krakow, Poland; eklimiec@ite.waw.pl (E.K.); pguzdek@ite.waw.pl (P.G.); kolaszczynski@ite.waw.pl (G.K.); jpiekarski@ite.waw.pl (J.P.); kzaraska@ite.waw.pl (K.Z.)

**Keywords:** foot, biomedical engineering, gait, pedobarography, piezoelectric sensors

## Abstract

Static and dynamic methods can be used to assess the way a foot is loaded. The research question is how the pressure on the feet would vary depending on walking/running speed. This study involved 20 healthy volunteers. Dynamic measurement of foot pressure was performed using the Ortopiezometr at normal, slow, and fast paces of walking. Obtained data underwent analysis in a “Steps” program. Based on the median, the power generated by the sensors during the entire stride period is the highest during a fast walk, whereas based on the average; a walk or slow walk prevails. During a fast walk, the difference between the mean and the median of the stride period is the smallest. Regardless of the pace of gait, the energy released per unit time does not depend on the paces of the volunteers’ gaits. Conclusions: Ortopiezometr is a feasible tool for the dynamic measurement of foot pressure. For investigations on walking motions, the plantar pressure analysis system, which uses the power generated on sensors installed in the insoles of shoes, is an alternative to force or energy measurements. Regardless of the pace of the walk, the amounts of pressure applied to the foot during step are similar among healthy volunteers.

## 1. Introduction

The main techniques used to assess the relationship between the structure of a foot and the biomechanics of its attached lower limb are kinematic tests (which examine the movements of individual body segments), electromyography, and analysis of the foot pressure on the ground (from both static and dynamic perspectives) [1]. These techniques are used by, among others, researchers looking for commonalities among different gait patterns [2].

Gait analysis is usually associated with a special room and apparatus. With this gait laboratory, very accurate measurements can be obtained, but usually under artificial conditions, e.g., on a flat surface and under observation. However, in an everyday environment, people naturally walk at different speeds; for example, a calm walk can be interpreted differently by different people. This is an important factor that should be considered in research concerning foot motion.

Static and dynamic methods can be used to assess the way a foot is loaded. The simplest static method is a pedobarographic examination, which requires standing in one position on a mat with sensors for a certain amount of time [3,4,5,6]. However, this study does not answer questions regarding how the foot walks. For this, dynamic methods must be used, where subjects are often instructed to walk on long mat “pavements” with installed sensors, and experiments are usually conducted in a gait laboratory [7,8]. Volunteers or patients had to walk with various speeds [8]. However, during these tests, there may be patients who, under special conditions, e.g., while on a mat with installed sensors, walk differently from their natural walking styles [9]. Therefore, it is important to be able to easily examine a patient on the conditions of natural gait, which is normally sometimes fast and sometimes slow [7,10].

The aforementioned methods of gait analysis sometimes also utilize sensors that are installed in the insole of the shoe [10,11,12]. However, similar sensors can be used in various items, e.g., seat covers for driver fatigue detection [13].

Therefore, the aim of this study is to present an analysis of foot pressure for people walking at different speeds, using the new Ortopiezometr. The question to be answered is: Does walking speed influence the energy generated from foot pressure on the ground as measured by sensors inserted into the shoe?

## 2. Materials and Methods

This study involved 20 volunteers. Volunteers reported to research after an announcement on the students’ intranet. The inclusion criteria for the study group were as follows: no foot injuries, no diseases and/or congenital anomalies, and no surgical and/or orthopedic treatments in relation to the foot. Flat foot and cavus foot were excluded as well. There were 20 healthy participants, students from Poland, aged between 20 and 25 years. They had normal BMI, their weight varied from 60 to 90 kg. Control of the correct pressure of the foot was implemented using a Tekscan pressure sensor mat (MatScan Clinical 6.62 program).

The study was approved by the local Bioethics Committee no. 122.6120.73.2015.

Dynamic measurement of foot pressure was performed using the Ortopiezometr (Polish Patent P.402006). Each measuring shoe insole contained 8 piezoelectric sensors, placed in anatomical zones according to Blomgren and Lorkowski [14,15] (Figure 1a,b). Volunteers used the same type of trainers (same model, only different sizes) to avoid any influence of the shoes on the gait pattern [16].

The measured data were transmitted wirelessly to a computer [17,18,19]. The measurements were performed in a 20-m long corridor, where each subject walked several times at normal, slow, and fast paces. The velocity of the fast/slow walk depended on each volunteer’s decision. The first three steps and last three steps during walking were excluded from evaluation due to unstable gait patterns. The authors had only one ortopiezometr system, so the data only from the one leg (right) could be collected.

Obtained data (for the right foot) underwent analysis in a “Steps” program (Visual Basic for Applications, MS Excel). The system logged the output voltage (in volts) for each of the 8 transducers; this voltage was proportional to the applied force (linear relationship). Subsequently, the energy *E* generated from the foot pressure on the ground during each walk was calculated:$$E\;=\;\sum_{i\;=\;1}^{n}(U_{i}^{2}s)\;=\;s\sum_{i\;=\;1}^{n}U_{i}^{2},$$
where *E* represents the energy, *U* the transducer voltage, *i* the sample number, and *s* the sampling period (15.5 ms).

For a quantitative comparison of walking, slow walking, and running, the generated power was calculated as follows:$$P\;=\;\frac{E}{t}\;=\;\frac{E}{ns}\;=\;\frac{s\sum_{i\;=\;1}^{n}U_{i}^{2}}{ns}\;=\;\frac{\sum_{i\;=\;1}^{n}U_{i}^{2}}{n},$$
where *P* represents the power, *n* the total number of samples, and *t* the total measurement time (*t*1, *t*2, *t*3, or *t*4, which are differentiated as follows). This way of presenting the results allowed their quantitative assessment: power (described as energy generated in transducers per time unit), taking into account not only the power of foot strike but also the duration of foot contact with the ground.

Each step of the foot can be divided into several phases. The first phase, i.e., between the heel’s contact with the surface and the moment when the heel pressure on the ground is zero, is labeled *t*1. The energy produced on the transducers during this phase divided by *t*1 is the average power for the “heel” phase. It equaled heel rocker according to Perry [20]. The second phase is from the moment when the heel pressure on the ground is zero (i.e., immediately following the end of the first phase) until the end of contact of the foot with the ground. This is referred to as the “front” phase. This phase corresponds to the ankle rocker and forefoot rocker phases according to Perry [20]. The third phase is a combination of the first two phases, i.e., the entire period of contact between the foot and the ground. Its duration is labeled as *t*3 = *t*1 + *t*2. This phase is referred to as the “stance” phase. The entire period between the first heel contact with the ground and the next heel contact is referred to as the “stride” phase and is labeled *t*4 (Figure 2).

The amounts of energy generated in the “stride” and “stance” phases are the same; however, because the duration of the “stride” phase is longer (*t*4 > *t*3), the corresponding power is lower.

To answer the research questions and test the hypotheses, statistical analyses were conducted using the IBM SPSS Statistics version 25 package. This software was used to analyze basic descriptive statistics and to perform the Friedman test. The classic threshold α = 0.05 was adopted as the level of statistical significance. In the first stage, basic descriptive statistics were calculated, and the Shapiro–Wilk test was applied to examine the normality of the distribution of variables measured on a quantitative scale. In the next stage of statistical analysis, the Friedman test was performed to check whether there were differences between the forces of foot pressure on the ground in the cases of running, normal gait, and slow walking. A post hoc Dunn–Bonferroni test was then used to analyze in detail the differences between the values measured for different gait rates.

## 3. Results

### 3.1. Basic Descriptive Statistics of Measured Quantitative Variables

Basic descriptive statistics were calculated first, and the distributions of all variables were determined to be close to a normal distribution (Appendix A).

For a more accessible approach toward the measurement data, the dependence of the average and median power for individual parts of the foot and the stride period on different modes of movement, and the relationship of the mean and median with different modes of movement, considering both the individual parts of the foot and the stride period, are shown in Figure 3, Figure 4, Figure 5 and Figure 6, respectively.

The power of the “back of the foot” is the greatest during a fast walk, and the lowest during a slow walk, whereas the power of the “front of the foot” is the lowest during a normal walk. When the median is considered, power during walking fast is biggest during heel and stance phases, whereas in front and stride phases differences between power noted in various walking paces are smaller.

Based on the median, the power generated by the sensors during the entire stride period is the highest during a fast walk, whereas based on the average, a walk or slow walk prevails. During a fast walk, the difference between the mean and the median of the stride period is the smallest.

### 3.2. Foot-to-Ground Pressure Applied with Respect to Pace of Walk

The next step was to conduct a Friedman test to examine if there were differences between the forces of the foot pressing on the ground in the cases of fast walking, normal walking, and slow walking. The test results show statistically significant differences: χ^2^(11) = 88.67; *p* < 0.001; *W* = 0.67.

Post hoc Dunn–Bonferroni tests revealed that in the case of a normal gait, the only notable differences were between the pressure force on the back of the foot and the pressure force in the stride period. The pressure on the entire foot was much less than on the back of the foot.

In contrast, for the running mode of movement, statistically significant differences were observed between the pressure force during the stride period and the power during loading of the hindfoot and in the stance phase of the foot. Thus, the pressure on the entire foot during the stride period was much weaker compared to the pressure on the back of the foot and in the stance phase. On the other hand, for the slow walk (stroll), the pressure force on the front of the foot was determined to be significantly greater than the force during the stride period. The values of the significance of these post hoc tests are presented in Table A2 (Appendix A). The mean ranks and medians for the variables are presented in Table A3 (Appendix A).

A comparison of normal walking, fast walking, and slow walking in terms of the pressure applied to the same part of the foot showed no statistically significant differences. Regardless of the pace of gait, the pressure on a given part of the foot is similar, i.e., the energy released per unit time does not depend on the pace of the volunteers’ gait. Table A4 in Appendix A lists the post hoc statistical significance values for these comparisons.

The differences depending on the different gait paces and pressure strengths for different parts of the foot were then analyzed. First, the amounts of pressure applied for a normal walking pace and the amounts of pressure applied to the various parts of the foot during a walk and a fast walk were compared. Table A5 (Appendix A) lists the post hoc significance values between a normal walk, and a fast walk and walk. In the next stage, various types of walking were compared depending on the pressure of the foot on the ground (Table A6 in Appendix A). The mean ranks obtained from the test and the medians of the analyzed variables are presented in Table A3 (Appendix A). The power dissipated on the sensors differed significantly between the individual phases of gait depending on the pace of the walk. However, these differences did not add up to a coherent whole, e.g., the pressure during the stride period of a walk differed significantly from that during the stance phase of a fast walk.

## 4. Discussion

The relationship between gait mechanics and the running ground-reaction forces is the subject of many studies worldwide [7,21]. Initial efforts were focused on trying to explain the rising edge of vertical force–time waveform phenomena. Solutions in the form of lumped-element spring–mass systems have appeared [21]. The problem is quite complex, however, because the total waveform corresponds to the acceleration of a body’s total mass (including trunk and limbs). The studies by Ly et al., and Zadpoor and Nikooyan described this part of the waveform quite well, but these models did not predict the falling edge of the waveform [22,23]. The varying shapes of the curves in this range probably depend on forefoot mechanics. Clark attempted to describe these dependencies at different speeds of movement by creating a two-mass-component model. In an experimental study with 42 volunteers, he examined the predicted model and real-time force–time waveforms [21]. For their research, they adopted a foot movement model similar to the 3-rocker model [20]. This model has worked well: the results show that the mass quantities and force–motion relationships do not differ across stride types. According to this model, waveforms can be predicted for walks at different speeds [21]. Thus, certain data can be standardized into a mathematical model regardless of the walking speed.

With regard to ground-reaction forces research, an important perspective to consider is that from the field of pedobarography. A pedobarograph is a tool that measures the pressure of the foot on the ground during standing or dynamic loading [24]. The obtained data allow the detection of disturbances in the foot structure and subsequent disturbances in its pressure [25]. Together with the insole creation program, it is a useful tool for reducing loads, e.g., in diabetic or flat feet [24,26].

The traditional mid-gait technique is the method used in gait laboratories: the patient walks across a walkway, and pressure data are collected from a single foot contact over a sensor platform. A better solution to eliminate targeting and attempts by the patient to modify their gait to hit the platform with their foot is to use insoles with pressure sensors in their shoes [9,11]. The data are obtained when the person is walking with shoes (not barefoot), which is, in fact, the usual way of walking. Moreover, it has been shown that walking barefoot and walking in shoes differ significantly [27].

Most pedobarographic studies assess changes in gait and/or foot load qualitatively or semi-quantitatively. It is true that peak or average pressure, force, and area are measured, but from the point of view of statics, the size of the sensor, resolution of the pressure measurement systems, and the possibility of easy data interpretation are all very important. Similarly, a relevant factor from the perspective of testing quality is the sampling frequency, which is measured in cycles per minute or in hertz [9]. When testing is performed using insoles, the resolution is usually lower, the sensors are more sensitive to mechanical damage, and the transducer cables (between the sensors and transducer) can be bent or stretched. In addition, a warm, humid environment adversely affects the condition of the device. Thus, the authors of this work implemented a slightly different, new solution for their experiment. The insert with Ortopiezometr sensors that we used is resistant to temperature (below 80 °C) and humidity, and the connecting cables are relatively short and resistant to mechanical damage.

Buldt emphasized that static foot posture disorders are a risk factor for lower limb injuries [1]. The foot structure itself is sometimes assessed in a pedobarographic examination either qualitatively or quantitatively [1,28,29]. A static examination describes well the anatomical abnormalities of the foot, e.g., a flat, hollow foot [30,31]. The person’s manner of walking during the test may vary; either there is no mention of the pace that should be used, or the patients are asked to walk or run at their own pace. In these tests, the walking speed is seldom clearly defined [1,32,33,34]. Other authors precisely established the walking speed [35].

The walking pace was not strictly measured in the study presented in this paper. Three gait speeds were also measured: normal, slower than normal, and faster than normal, similarly to Andriacchi or Fernandes [7,36]. The observations from this study indicate that changes in the power output on the sensors vary with the pace of walking, but specific patterns were not detected among these variables.

The forces and work performed by the foot and ankle while walking can be difficult to analyze, especially in terms of a breakdown of the individual parts of the foot [37]. For this purpose, multi-segment models can be used to describe the movement of individual parts of the foot, i.e., kinematics [38,39,40]. These foot models based on markers on the skin of the foot represent the movement of the footwell and are routinely used in gait laboratories [41]. However, the kinetics of the foot, as a description of forces, power, and work, remains a challenge [37]. The multi-segment foot model seems to describe these phenomena better compared to the one-segment model [42,43,44,45]. However, there is no single valid model, because contemporary methods of dynamic rate assessment differ on many levels. Even the dimensions that are measured among these models are different; the most commonly used is the peak plantar pressure, which describes the maximum pressure value on a specific area of the foot during the gait cycle [1]. The units in terms of which pressure is measured are also different; sometimes, the unit used is N/cm^2^, or kPa, or may not even be provided in the documentation [28,32]. Another value that is sometimes measured during the analysis of foot loading is the maximum force recorded for a certain area it does not depend on the magnitude of that area. An increased force was noted by Hillstrom at the big toe for a hollow foot, whereas Chuckpaiwong described a lower lateral force on a flat foot compared to that on a normal foot [28,46]. Most of these studies describe the change in maximum force in terms of foot/knee pathology compared to that of healthy subjects [36]. These are comparative studies, and thus do not provide a standard range so the question of the objective range of correct values at different speeds occurs. An attempt to answer this question is the evaluation of the force–time integral, measured as the cumulative exposure to force over time on a part of the foot [28,32,46].

It seems that the definition of plantar pressure used in this study, based on the measurement of power as the value of energy exerted on a sensor per unit of time, is a means of obtaining objective results that can be compared among individual patients and at different walking paces. The amount of power in the range of the whole stride may have been the greatest during a fast walk, but a slow walk also produced high measurements. This can be explained on the one hand by the large amount of energy released in a short time during a fast walk and a very slow walk; although the time was longer for the latter, the part of the step when the foot rested with its entire surface on the ground generated considerable energy.

During a normal walk, the power generated by the sensors on the back of the foot is lower than the power corresponding to the pressure on the entire foot, because the load is exerted only on the heel for a short amount of time, during which the load exerted is the entire body weight. This is the case when a person walks properly. On the other hand, walking with an incomplete load on the heel, in the case of a plantar contracture of the foot, should generate a different image; this is an area for further research. We know, that gait speed can influence (somehow) the stability of walking and gait pattern [12,47,48,49,50].

In this study, it was difficult to interpret the differences in the power of the back of the foot and of the entire foot when the subjects were walking fast. Authors suspected running instead of fast walking. However, the pressure on the back of the foot was greater, which should not be the case during a run (unless the amateur runs were not toe runs). This gait pattern exactly matches a very fast walk. It corresponds with the results of Breine, who emphasizes the different ways of running among runners [51].

The analysis of the relationship between the different gait phases and the different gait rates did not provide clear results. Perhaps the lengths or durations of the walks during the study, or the number of volunteers, were not sufficient for obtaining some pattern of power output from the sensors. The observations in this study are consistent with those from the studies by Hellstrom, who noticed certain dependencies in the results at different walking [52].

We think that the future of foot pressure research is related to the analysis of a natural, comfortable gait, but due to variability of peak plantar pressure at different walking speeds, detailed measurement of speed may be necessary [35,53,54,55].

The limitations of the study are certainly the small number of volunteers and the relatively short 20 m measuring section and not measured walking speed.

## 5. Conclusions

This research, therefore, arrives at the following conclusions:Ortopiezometr is a feasible tool for dynamic measurement of foot pressureFor investigations on walking motions, the plantar pressure analysis system, which uses the power generated on sensors installed in the insoles of shoes, is an alternative to force or energy measurements;Regardless of the pace of the walk, the amounts of pressure applied to the foot during step are similar among healthy volunteers;Further research is needed to investigate feet with abnormal anatomies.

## Figures and Tables

**Figure 1 sensors-22-03098-f001:**
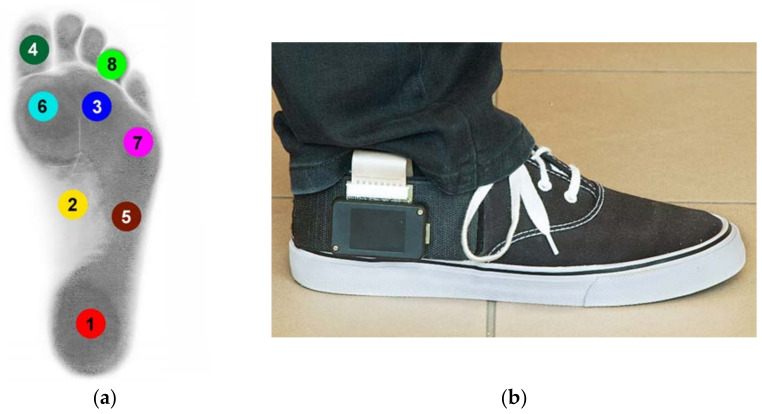
(**a**) Placement of 8 piezoelectric sensors in the shoe insole. (**b**) Ortopiezometr—transducer attached to the shoe.

**Figure 2 sensors-22-03098-f002:**
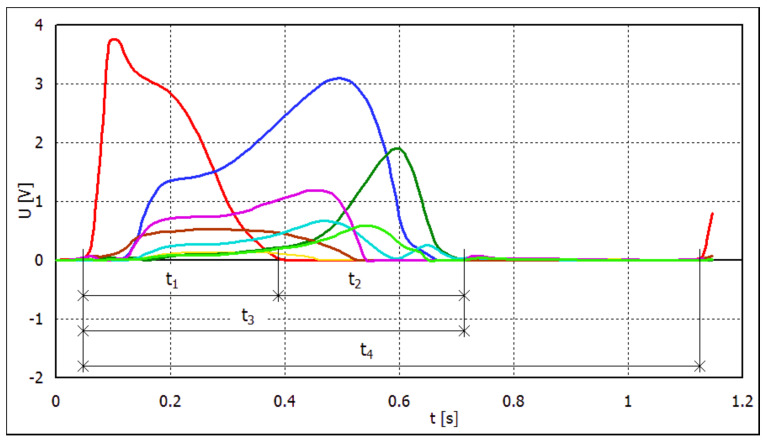
Phases: *t*1, *t*2, *t*3, and *t*4 of the step.

**Figure 3 sensors-22-03098-f003:**
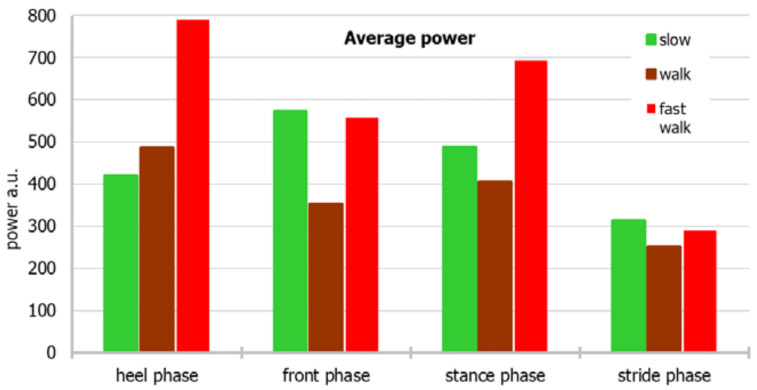
An average power considering both the individual parts of the foot and the stride period.

**Figure 4 sensors-22-03098-f004:**
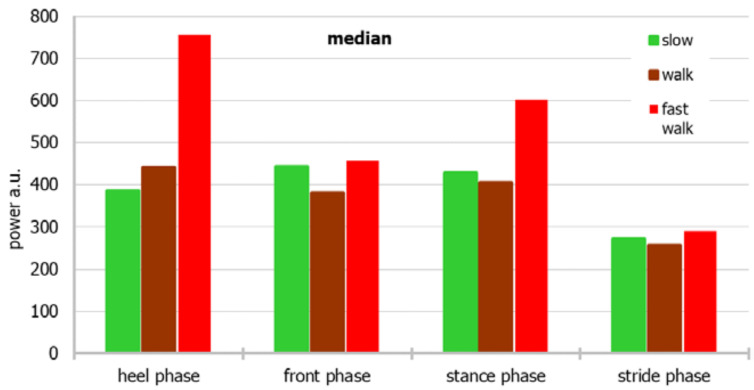
The median power for individual parts of the foot and the stride period on different modes of movement.

**Figure 5 sensors-22-03098-f005:**
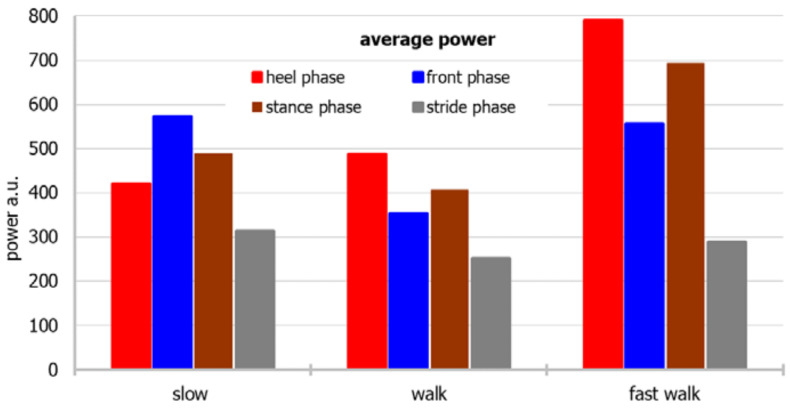
Mean power according to different modes of movement.

**Figure 6 sensors-22-03098-f006:**
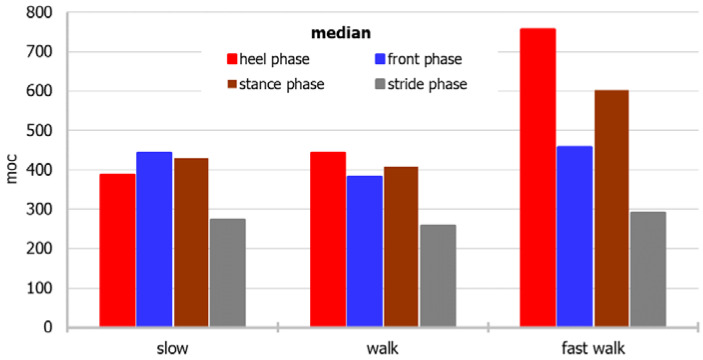
Median power according to different modes of movement.

## Data Availability

Raw data from orthopiezometr are available from authors upon request.

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
