# Peer review of "Investigation of Impact of Walking Speed on Forces Acting on a Foot–Ground Unit"

_sensors, 2022, doi:10.3390/s22083098_

Round 1

Reviewer 1 Report

In this paper, the Authors are proposing to examine how the pressure on the feet would vary depending on walking / running speed.

They involved 20 volunteers, control of the correct pressure of the foot was implemented using a Tekscan pressure sensor mat.

After carefully reading, I find that this paper is extremely interesting, and the results very promising, however in order to further improve I would only recommend to improve the conclusions and more references on the background. (I suggest: doi: 10.3390/app112211020, doi: 10.3390/sym13081476, doi: 10.3390/e23020135, doi: 10.3390/s21093217)

Author Response

Thank you for your comment. As you propose, we tried to improve conclusions and we added more references:

Lorkowski J, Gawronska K, Pokorski M. Pedobarography: A Review on Methods and Practical Use in Foot Disorders. Applied Sciences. 2021; 11(22):11020. https://doi.org/10.3390/app112211020

Gawronska K, Lorkowski J. Evaluating the Symmetry in Plantar Pressure Distribution under the Toes during Standing in a Postural Pedobarographic Examination. Symmetry. 2021; 13(8):1476. https://doi.org/10.3390/sym13081476

Salvati L, d’Amore M, Fiorentino A, Pellegrino A, Sena P, Villecco F. On-Road Detection of Driver Fatigue and Drowsiness during Medium-Distance Journeys. Entropy. 2021; 23(2):135. https://doi.org/10.3390/e23020135

Zhang G, Wong DW-C, Wong IK-K, Chen TL-W, Hong TT-H, Peng Y, Wang Y, Tan Q, Zhang M. Plantar Pressure Variability and Asymmetry in Elderly Performing 60-Minute Treadmill Brisk-Walking: Paving the Way towards Fatigue-Induced Instability Assessment Using Wearable In-Shoe Pressure Sensors. Sensors. 2021; 21(9):3217. https://doi.org/10.3390/s21093217

Also all references were checked, and re-numbered.

Reviewer 2 Report

The paper can be of interest for the journal’s readers, but some issues must be resolved before its publication:

  1. Data from which steps were taken for the analysis? The subjects walked at the 20 m long corridor, so there were both acceleration (beginning) and deceleration (end) phases, in which the gait pattern is not stable. It is assumed that the first two-three steps and last two-three steps should be discarded to this fact.
  2. Was the anatomy of the subjects’ feet checked before the enrolment of the study? Pes planus or pes cavus could seriously modify the load distribution while walking and running.
  3. Terminology – the authors should use terms well established in the literature dealing within the gait analysis field (eg. two books: by R.Baker – A handbook of clinical gait analysis:, or D.levine, J.Richards, MW Whittle – Whittle’s gait Analysis):
  • Lines 102-103 – the third phase (“the entire period of contact between the foot and the ground”) which the authors called “support” phase should be called “stance phase”
  • Lines 105-106 – the “step” phase, i.e. from the heel contact to the heel contact of the same foot is “gait cycle” or “stride” (by the way, the authors themselves use term “stride” later in the paper – line 130).
  1. Shoes can modify in many ways the gait pattern and loading of the foot the information about the type of shoes used by the subjects is necessary (eg. Park & Park, J Exrc Rehabil 2018, Arnadottir & Mercer Phys Therapy 2000). The possible influence of the shoes on the results should be discussed.
  2. The description of the test used for checking the type of variables’ distribution should be moved to the Methods (lines 124-126).
  3. There is no rationale for using only the data from the right leg (line 79), and not, first comparing left and right leg data pooling them together for further analyses. Maybe the authors had only one system, and the data from the one leg could be collected, it is ok, but must be clearly stated.
  4. In line 81 the authors stated that the voltage in the units is proportional to the applied force. Was the relationship linear? If not some additional calculations should be done.
  5. Another problem is the force applied to the sensors: it depends not only on the way in which the foot is loaded during the stance phase, but mainly on the body weight of the subject. We know nothing about the body dimensions and body weight of the subjects, but even if they were more or less homogenous the normalization on the body weight of the data should be done.
  6. The sentence in lines 142-144 is completely unclear for me: what means the word “dominant” in this context? Either someone is running or not, and it she/he runs this is the only type of activity performed. The word “advantageous” is an evaluative word, what is better, and what is worse should be placed in the Discussion section, with described criteria.

Author Response

Thank you for your comment. We did our best and tried to improve the text according to them.

Data from which steps were taken for the analysis?

the first three steps and last three steps during walking were discarded (not stable gait pattern)

Was the anatomy of the subjects’ feet checked before the enrolment of the study?

Foot anatomy of volunteers was normal (Pes planus or pes cavus were excluded from the study, as well as feet with previous surgeries).

Terminology – the authors should use terms well established in the literature dealing within the gait analysis field (eg. two books: by R.Baker – A handbook of clinical gait analysis:, or D.levine, J.Richards, MW Whittle – Whittle’s gait Analysis):

Terminology was changed to proper acc. to above mentioned. Also in tables and figures.

Shoes can modify in many ways the gait pattern and loading of the foot the information about the type of shoes used by the subjects is necessary

Volunteers used the same type of trainers (same model, only different size) to avoid any influence of the shoes on the gait pattern. Suitable ref. was added.

The description of the test used for checking the type of variables’ distribution should be moved to the Methods (lines 124-126).

It was done.

There is no rationale for using only the data from the right leg (line 79), and not, first comparing left and right leg data pooling them together for further analyses. Maybe the authors had only one system, and the data from the one leg could be collected, it is ok, but must be clearly stated.

Authors had only one ortopiezometr system, so the data only from the one leg could be collected.

In line 81 the authors stated that the voltage in the units is proportional to the applied force. Was the relationship linear?

The relationships is linear, detailed data available on request.

Another problem is the force applied to the sensors: it depends not only on the way in which the foot is loaded during the stance phase, but mainly on the body weight of the subject. We know nothing about the body dimensions and body weight of the subjects, but even if they were more or less homogenous the normalization on the body weight of the data should be done.

Volunteers  had normal BMI, their weight varied from 60 to 90kg.

The sentence in lines 142-144 is completely unclear for me: what means the word “dominant” in this context? Either someone is running or not, and it she/he runs this is the only type of activity performed. The word “advantageous” is an evaluative word, what is better, and what is worse should be placed in the Discussion section, with described criteria.

The sentence was changed,  I hope it is more clear now:

“When the median is considered, power during running is biggest during heel and support phases, whereas in front and step phases differences between power noted in various walking paces are smaller”. 

Round 2

Reviewer 2 Report

The authors addressed all the comments and introduced changes in the manuscript. Still there is one issue regarding the terminology. In the title there is "impact of walking speed", in the the study aim "does walking speed influence...", and in the graphs, results and discussion the locomotion with  highest speed is descrbed as "run". Although the authors stated, that (line 85) "Fast pace was considered equal to running" running and walking are two different types of activity. There is a range of speeds in which we can either walk or run, but during walking always at least one leg has contact with ground, and in healthy people the initial contact startc with heel. During running there is a phase when there is no contact of the groud and initial contact can be done (correctly) with forefoot. This issue should be better described before the publication of the paper. If the subjects really only walked I suggest using "fast walk" instead of "run". If some of tchem rwally run the data analysis has to be done again taking into account this fact.

Author Response

Thank you for your comments.

After analyzing the raw data, we agreed with your suggestion: our volunteers did not run , the walked as fast as possible (without a phase when there is no contact of the ground and initial contact can be done (correctly) with forefoot.

So, we changed “run” into “fast walk” in the manuscript.